**www.cambridge.org/ext**

## Overview Review

conservation palaeobiology; extinctions; extirpations; Quaternary; marine habitats

**Corresponding author:**
Michał Kowalewski;
Email: mkowalewski@flmnh.ufl.edu

# Marine conservation palaeobiology: What does the late Quaternary fossil record tell us about modern-day extinctions and biodiversity threats?

Michał Kowalewski[1] 📧, Rafał Nawrot[2] 📧, Daniele Scarponi[3], Adam Tomašových[4] and Martin Zuschin[2]

[1]Florida Museum of Natural History, University of Florida, Gainesville, FL, USA; [2]Department of Palaeontology, University of Vienna, Vienna, Austria; [3]Dipartimento di Scienze Biologiche, Geologiche e Ambientali, University of Bologna, Bologna, Italy and [4]Earth Science Institute, Slovak Academy of Sciences, Bratislava, Slovakia

## Abstract

Near-time conservation palaeobiology uses palaeontological, archaeological and other geo-historical records to study the late Quaternary transition of the biosphere from its pristine past to its present-day, human-altered state. Given the scarcity of data on recent extinctions in the oceans, geohistorical records are critical for documenting human-driven extinctions and extinction threats in the marine realm. The historical perspective can provide two key insights. First, geohistorical records archive the state of pre-industrial oceans at local, regional and global scales, thus enabling the detection of recent extinctions and extirpations as well as shifts in species distribution, abundance, body size and ecosystem function. Second, we can untangle the contributions of natural and anthropogenic processes by documenting centennial-to-millennial changes in the composition and diversity of marine ecosystems before and after the onset of major human impacts. This long-term perspective identifies recently emerging patterns and processes that are unprecedented, thus allowing us to better assess human threats to marine biodiversity. Although global-scale extinctions are not well documented for brackish and marine invertebrates, geohistorical studies point to numerous extirpations, declines in ecosystem functions, increases in range fragmentation and dwindling abundance of previously widespread species, indicating that marine ecosystems are accumulating a human-driven extinction debt.

## Impact statement

Whereas only a few marine species have gone globally extinct due to human activities, an increasing number of ocean-dwelling lifeforms are on decline. However, most scientific surveys and monitoring efforts only cover the last several decades and are thus insufficient to fully assess long-term human impacts on the marine biosphere. The late Quaternary fossil record and other geohistorical archives fill this gap by documenting marine biodiversity losses that have already taken place, pinpointing ecological shifts that exceed natural variability and improving our ability to identify species facing extinction threats. Whereas data and strategies focusing on present-day biodiversity will remain the critical dimension of conservation and ecosystem management, geohistorical approaches can augment those efforts by documenting biodiversity losses and threats that would not and could not have been discovered otherwise and by providing direct insights into the transition of the pre-human biosphere into its current state.

## Introduction

Extinctions and extinction threats linked to human activities are on the rise, prompting warnings about the arrival of the sixth mass extinction (e.g., Leakey, 1996; Kolbert, 2014; Ceballos et al., 2015; Régnier et al., 2015; Plotnick et al., 2016; Dasgupta and Ehrlich, 2019; Cowie et al., 2022). However, the current biodiversity crisis has deep historical roots (e.g., Soulé, 1985; Wilson, 1985; Jackson et al., 2001; Pandolfi et al., 2003; Lotze et al., 2006; Frank, 2022) that extend back far beyond the temporal span of the modern instrumental records, bio-inventorying efforts and long-term ecological monitoring programs, especially in marine ecosystems (see Figure 1 in Kosnik and Kowalewski, 2016). While data on present-day ecosystems are invaluable (e.g., Strong et al., 1997; Franke et al., 2004; Vinod et al., 2020), they mostly cover the last few decades (Vellend et al., 2013; Dornelas et al., 2014; Blowes et al., 2019) and their predominantly local spatial focus translates into patchy global coverage. In short, neontological data inform us about the most

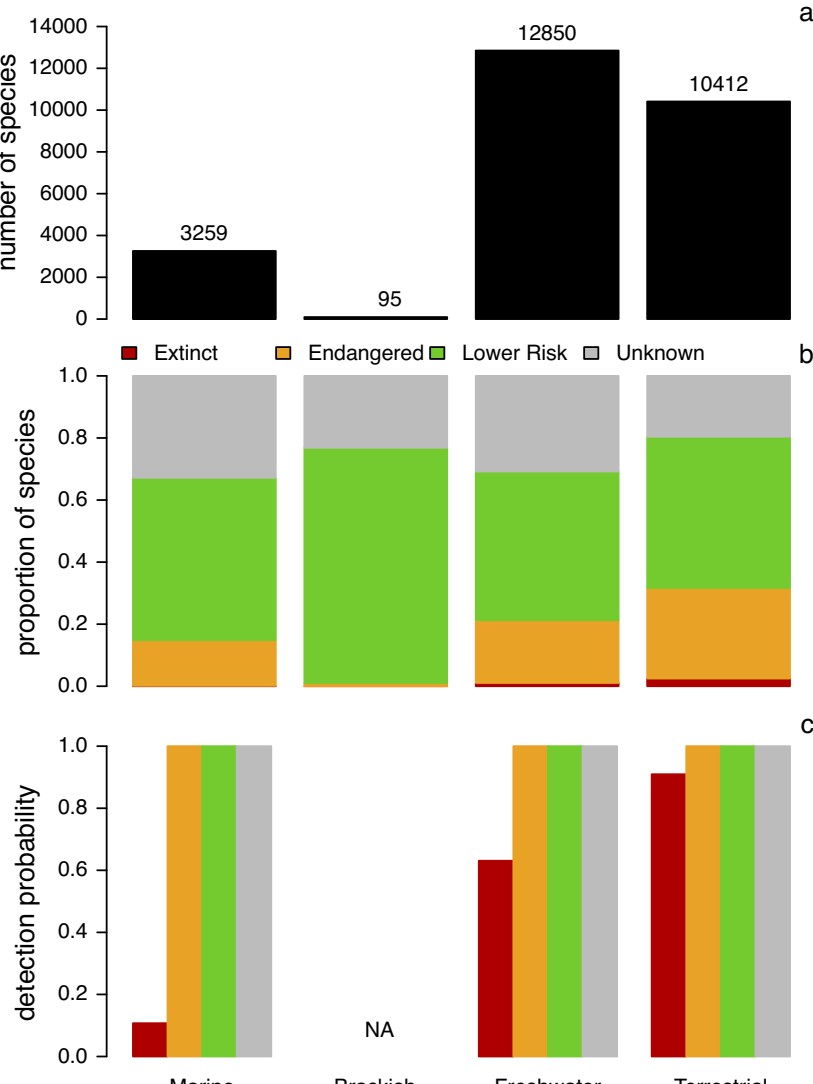

**Figure 1.** Comparison of four major invertebrate phyla (arthropods, cnidarians, echinoderms and molluscs) across marine, brackish, freshwater, and terrestrial systems as recorded in the IUCN Red List Database (IUCN, 2023a). (a) The total number of species reported in the database for each of the four systems (numbers indicate the total number of species). (b) The Red List status of invertebrate species grouped into four broad categories ('Extinct', 'Endangered', 'Lower Risk', and 'Unknown') tallied separately for each system. The broad categories were derived by pooling IUCN categories as follows: (1) "Unknown" – "Data Deficient"; (2) "Extinct" – "Extinct", "Extinct in the Wild"; (3) "Endangered" – "Critically Endangered", "Endangered", "Vulnerable"; and (4) "Lower Risk" – "Near Threatened", "Lower Risk/near threatened", "Lower Risk/conservation dependent", "Lower Risk/least concern", "Least Concern". See Supplementary Material for additional information.

recent eco-environmental changes in selected regions, even though those changes may have been ongoing for much longer and affecting other regions of the world.

Placing neontological observations in the context of the long-term dynamics of ecosystems is a prerequisite for assessing human impacts on our biosphere (Pandolfi et al., 2020). Conservation palaeobiology is an emerging geohistorical approach developed towards this goal (e.g., Flessa, 2002; Kowalewski, 2004; Froyd and Willis, 2008; Dietl and Flessa, 2011, 2020; Tyler and Schneider, 2018; Kiessling et al., 2019; Turvey and Saupe, 2019; Dillon et al., 2022; Nawrot et al., 2023). By using geohistorical archives (e.g., sediment cores, surficial skeletal assemblages, archaeological middens, geochemical proxies, ancient DNA) from the most recent centuries and millennia, conservation palaeobiology aims to document temporal trajectories in eco-environmental patterns, assess the timing, magnitude and forcing of past ecosystem

changes and elucidate processes associated with the transition of the pre-human biosphere to its current state. In addition, conservation palaeobiology provides a direct strategy for assessing deeper historical roots that underlie current extinction threats (e.g., habitat loss and fragmentation, range contractions, population declines) and may ultimately result in future extinctions (Saupe et al., 2019).

Over the last several decades, geohistorical approaches have been applied to study human impacts on terrestrial (e.g., Gorham et al., 2001; Behrensmeyer and Miller, 2012; Wood et al., 2012; Lyons et al., 2016; Barnosky et al., 2017; Jackson et al., 2017; Koch et al., 2017; Terry, 2018; Smith et al., 2022), freshwater (e.g., Brown et al., 2005; Smol, 2008; Erthal et al., 2011; Kusnerik et al., 2022; Czaja et al., 2023) and marine (e.g., Kowalewski et al., 2000, 2015; Jackson et al., 2001; Kidwell, 2007; Aronson, 2009; Tomašových and Kidwell, 2017; Hyman et al., 2019;

Cybulski et al., 2020; Albano et al., 2021; Dillon et al., 2021; Hong et al., 2021; Rivadeneira and Nielsen, 2022; Meadows et al., 2023; Scarponi et al., 2023) ecosystems. The above citations are only a fraction of novel studies aimed at establishing pre-Anthropocene baselines or improving the management and restoration of natural habitats by employing geohistorical data. Here, we focus on documenting how geohistorical approaches can improve our understanding of extinctions and extinction threats in the marine realm.

Most of the recent extinctions have been documented for terrestrial organisms, which are the main focus of extinction studies, while relatively less is known about other taxa, especially marine invertebrates (McKinney, 1998; Régnier et al., 2015; but see Cowie et al., 2022). Nonetheless, despite a multitude of anthropogenic impacts such as deoxygenation, heat stress, acidification, overfishing and pollution that led to significant range contractions and extirpations at regional scales (Scheffer et al., 2005; Jackson, 2008; Pusceddu et al., 2014), only a few species of marine organisms have been deemed extinct (del Monte-Luna et al., 2009; Dulvy et al., 2009; Régnier et al., 2015; Briggs, 2017; Cowie et al., 2022; del Monte-Luna et al., 2023). Nevertheless, many marine ecosystems have been degraded, and many marine organisms face extinction threats (e.g., Edgar et al., 2005; Lotze et al., 2011; McCauley et al., 2015; Penn and Deutsch, 2022). Consequently, the relatively low number of marine extinctions may reflect the dearth of conservation assessments for marine species, especially when compared to the notably more extensively studied terrestrial organisms, especially vertebrates (Webb and Mindel, 2015; but see Cowie et al., 2022).

A comparison of four widespread and diverse invertebrate phyla (Arthropoda, Cnidaria, Echinodermata, and Mollusca) listed in the International Union for Conservation of Nature's (IUCN) Red List database (IUCN, 2023a) is consistent with previous studies (e.g., Webb and Mindel, 2015) in demonstrating that terrestrial and freshwater invertebrates (arthropods and molluscs) are represented much more comprehensively than their marine and brackish counterparts (Figure 1a). In fact, only one marine and no brackish invertebrate species are reported as having gone extinct after 1,500 CE (the year IUCN uses as a cut-off for listing species as extinct; Figure 1b). However, consistent with the recent literature, the database identifies numerous marine invertebrates as endangered, indicating that our neontological knowledge of marine extinction threats is growing. Here, we will consider how geohistorical archives can augment our understanding of marine extinctions and extinction threats.

One obvious caveat applies to the IUCN analysis above. The IUCN sampling coverage varies greatly across systems, and thus, the absence of extinctions in the systems with a more limited IUCN assessment may reflect undersampling. In particular, the brackish system (n = 95 species) is poorly sampled. When other systems are sample standardised to the sample size of the smallest system (n = 95 species), the probability of detecting at least one extinction event is relatively high for freshwater systems (~1% of the assessed species classified as extinct) and high for terrestrial systems (~2.5% of the assessed species classified as extinct) but low for the marine system (~0.1% assessed species classified as extinct). The probabilities of detecting at least one extinction are 0.63, 0.91 and 0.11, respectively (see Supplementary Material). A total of 95 species should be enough to detect extinct species in brackish systems (detection probability >0.99) only if extinction rates exceeded 5%. This caveat itself involves a caveat, however. The above estimates rely on an assumption that assessed species are a random sample of all species in a system.

## Extinction terminology

To minimise terminological ambiguity, we provide explicit definitions of extinction types. In its strictest formal definition, extinction is typically understood as a complete and irreversible disappearance of a species on a global scale. However, geographic range contractions, population declines or shifts in functional traits can undermine ecosystem health and services as much as global extinctions. Past mass extinctions may have been mass rarity events, with rarity being practically equivalent to extinctions, in terms of both ecological consequences as well as the resulting fossil record of biodiversity (Hull et al., 2015). Building on terminology reviewed in previous studies (Estes et al., 1989; Carlton et al., 1999; McConkey and O'Farrill, 2015; McCauley et al., 2015), we distinguish here three main categories of extinctions.

*Extinction* – A total disappearance of a species. Also referred to as "global extinction" (Estes et al., 1989).

*Extirpation* – A local or regional disappearance of a species still occurring elsewhere ("local extinction" sensu Estes et al., 1989). Extirpations can lead to the fragmentation of geographic ranges and range contractions. However, not all extirpations lead to the decline in geographic range extent. For example, human harvesting of large limpets such as *Scutellastra mexicana* resulted in the demise of many (but not all) local populations along Mexico's coast (Carballo et al., 2020). Consequently, the northern latitudinal range of this species has not contracted notably despite those numerous extirpations. Similarly, the latitudinal range of the iconic marine mammal (*Dugong dugon*) was not reduced by its human-driven extirpation from the Spermonde Archipelago in central-western Sulawesi (Moore et al., 2017).

*Ecological Extinction* – Ecological decline of a species that is still present but very rare and no longer plays a significant ecological function or interacts significantly with other species (McConkey and O'Farrill, 2015). Ecological extinctions may lead to the extinctions of other species in the community (Säterberg et al., 2013). Ecological extinction is usually driven by "decimation", a dramatic decline in population density. Such decline can also lead to a drop below an abundance level at which a species can be economically harvested, referred to as "commercial extinction" (Carlton et al., 1999; McCauley et al., 2015). A significant decline in abundance tends to correlate with range contractions (Worm and Tittensor, 2011). Ecological extinction can also be driven by changes in functional traits of a species. For example, the loss of larger size classes and older age cohorts can diminish the role a species plays in the ecosystem (e.g., Norkko et al., 2013; Hočevar and Kuparinen, 2021). In many studies, the term "functional extinction" is equivalent to "ecological extinction" (e.g., McConkey and O'Farrill, 2015; Ebenman et al., 2017), but it has also been used to denote a permanent lack of reproductive or recruitment success (see Jarić et al., 2016 and references therein).

*Extinction Debt* – This conceptual addendum to the extinction terminology posits that biodiversity loss lags anthropogenic environmental pressures (e.g., habitat fragmentation; Tilman et al., 1994; but see MacArthur, 1972). Human-driven ecosystem perturbations can increase the probability of species extinction and induce "extinction debts" when population sizes decline below their functioning thresholds (Malanson, 2008). However, intrinsic species traits (e.g., dispersal capacity, longevity, genetic plasticity), (meta)

population dynamics (e.g., degree of connectivity) and species interactions can delay species disappearance. Thus, ecosystems tend to accumulate extinction debts during and after a perturbation (Hanski and Ovaskainen, 2002). As species go extinct, the "debt" is progressively paid off, and ecosystems shift towards a new equilibrium state. Although neontological data suggest that the time needed to pay off the debt, known as "relaxation time", can range from only a few years to several centuries (Forman and Godron, 1986), the fossil record demonstrates that a lag between environmental perturbations and resulting extinctions can reach even 2 million years (O'Dea et al., 2007; O'Dea and Jackson, 2009). The magnitude of the debt in an ecosystem depends on the number of affected species. Whereas both theoretical and empirical approaches for detecting extinction debts have been developed (Kuussaari et al., 2009; Figueiredo et al., 2019), quantifying extinction debts and relaxation times has proven challenging. The extinction-debt investigations have focused on continental settings, and only a few studies have dealt with marine ecosystems (see Briggs, 2011). Yet, the concept of extinction debt is valuable in its potential to identify extinction threats, forecast future extinctions and assess the common drivers (the trifecta of habitat destruction, climate change and invasive species).

The concept of extinction debt may be spatially and temporally scalable (e.g., "mass extinction debt" concept in Spalding and Hull, 2021), allowing to use geohistorical data for predicting future extinction risks. Palaeontological and other geohistorical investigations can act synergistically with ecological monitoring or theoretical models by providing historical estimates of the onset of ecosystem decline and magnitude of extinction debt that accumulated in the past. Documenting when the ecosystem decline started and estimating the extent of losses that have already occurred can help to evaluate relaxation times more precisely. Finally, documenting historical changes in ecosystems, which often reflect responses to natural (non-anthropogenic) processes, can also allow us to disentangle natural and anthropogenic drivers of extinctions.

## Geohistorical perspectives on extinctions and extinction threats

The value of geohistorical approach resides in its potential to detect extinctions, extirpations and ecosystem changes not discoverable by neontological data. For example, modern biomonitoring data focused on a particular clade may indicate that one of the two species disappeared (50% extinction), but the fossil record may demonstrate that three additional species existed in pre-industrial times. Thus, the within-clade extinction magnitude may be 80% rather than 50% of the species (Figure 2a). Seabirds inhabiting oceanic islands are a good example of underestimating human-driven extinctions, with more than 20 species lost worldwide during the Holocene (Tyrberg, 2009; Ramirez et al., 2010). For instance, out of four shearwater species (genus *Puffinus*) breeding on the Canary Islands, two went extinct during the Holocene (Ramirez et al., 2010; Rando and Alcover, 2010). However, these extinctions are not included in the IUCN Red List database, which uses 1500 CE as the cut-off year, even though those extinctions were linked to aboriginal colonisation of the archipelago and the introduction of exotic species following European settlement in the fourteenth century (Rando and Alcover, 2008, 2010). More generally, birds inhabiting islands, including both marine and land species, represent a remarkable example of how geohistorical data can transform our understanding of the timing and magnitude of human-driven extinctions. Based on zooarchaeological data, it has been estimated that as many as 2,000 bird species from Pacific tropical islands may

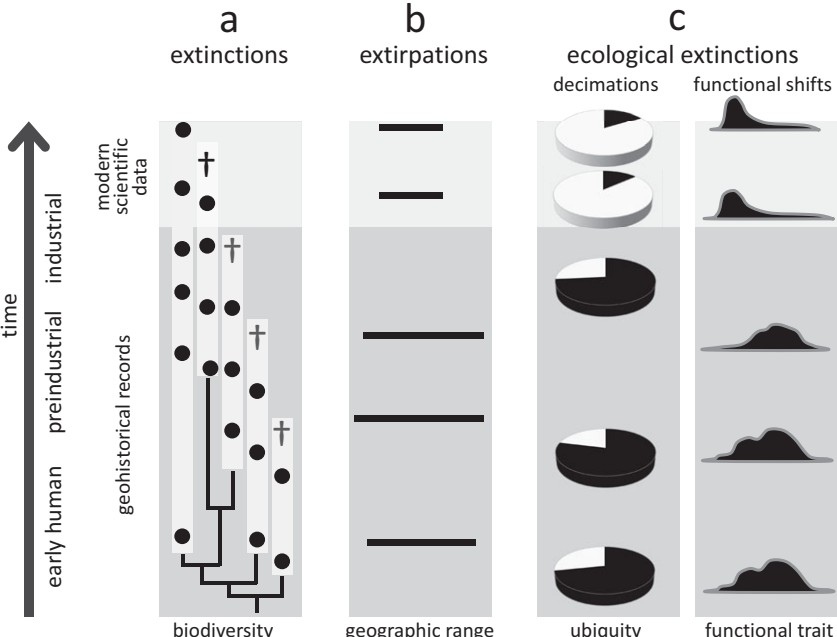

**Figure 2.** Conceptual illustration of how geohistorical data augment our understanding of global extinctions, extirpations, and ecological extinctions. In all examples, geohistorical knowledge indicates significant losses that would not be discoverable using modern scientific data alone. (a) Extinctions - an example of a clade for which the extinction rate is much higher once geohistorical data are considered (see text for a case example of the seabirds *Puffinus*). White bars indicate estimated stratigraphic ranges based on fossil occurrences (black circles), crosses indicate terminal extinction events; (b) Extirpations - an example of range contraction that becomes apparent only after geohistorical data are included (see text for case examples of seals and marine molluscs); (c) Ecological extinctions - examples of decimations (declines in population density) and functional shifts (e.g., shortened life spans, diminished body size) that become apparent once the fossil record is considered (see text for a case example of *Crassostrea virginica* in Chesapeake Bay, USA).

have been driven to extinction by prehistoric human activities (Steadman, 1995, 2006).

Similarly, archaeological and palaeontological data have proven useful in documenting past extirpation events and historical shifts in geographic ranges of marine taxa, providing evidence for the presence of aquatic species in regions from which they disappeared long before we started to collect modern scientific data (Figure 2b). For example, palaeobiological records from the Baltic Sea documented biogeographic shifts in several species of marine mammals during the Holocene (Sommer and Benecke, 2003). One of these species, the harp seal (*Pagophilus groenlandicus*), has been extirpated twice from the area (at the end of the Middle Holocene and then again during the Medieval Warm Period), and, in both cases, human activities and climate changes may have been contributing factors (Glykou et al., 2021). These inferences were based on integrated analyses of zooarchaeological, palaeoecological, radiometric and geochemical data (Glykou et al., 2021). Similarly, radiocarbon-dated bones suggest that gentoo and chinstrap penguins expanded their breeding distribution southwards in the Antarctic peninsula within the past several decades (Emslie et al., 1998). In contrast, the Adélie penguin has occupied the area for centuries, including many currently abandoned colonies. Still, the presence of this species may have been intermittent, possibly due to climatic fluctuations related to the Little Ice Age (Emslie et al., 1998). The extirpation of Adélie penguin colonies and the expansion of gentoo and chinstrap penguins may have been related to rapid regional warming (Emslie et al., 1998), likely linked to global climate changes. These types of historical records are even more readily available for marine invertebrates. For example, the analysis of mollusc shell assemblages from radiometrically dated sediment cores revealed that the commercially important oyster *Ostrea edulis*, which was once dominant along Scottish coasts, disappeared from the Firth of Forth in the nineteenth century due to bottom trawling (Thurstan et al., 2013). Those examples document extirpations that were likely linked to human activities, but many would have remained undetected without geohistorical data.

The historical perspective is particularly valuable in the case of ecological extinctions and resulting shifts in ecosystem functioning, which may be both pervasive and underreported in the marine realm (McCauley et al., 2015). In such cases, geohistorical data can correct modern perceptions of changes in the distribution and functional role of species (Figure 2c) and facilitate identifying extinction threats. For example, palaeoecological, historical and modern survey data demonstrated that the decline of Caribbean acroporid corals began in the 1950s, two decades before the onset of systematic monitoring efforts in the region (Cramer et al., 2020). Extensive U-Th dating indicated that acroporid corals from the Great Barrier Reef also had started declining before monitoring efforts were initiated (Clark et al., 2017). These efforts allowed for establishing more reliable baselines for future monitoring (Clark et al., 2017). Similarly, comparisons of modern and Pleistocene populations of the oyster *Crassostrea virginica* in Chesapeake Bay sampled from comparable environments revealed that this species could live much longer, grow to significantly larger sizes and achieve higher population densities than previously recognised based on monitoring surveys (Kusnerik et al., 2018; Lockwood and Mann, 2019). Without considering the Pleistocene fossil record, the magnitude of recent changes in lifespan and population structure of this species, attributed to the preferential harvest of larger oysters and disease-related die-offs (Andrews, 1996; Lockwood and Mann, 2019), would have been underestimated. These changes in the functional ecology of oyster reefs affected their filtering capacity, estimated to be an order of magnitude greater in the past (Lockwood and Mann, 2019), and thus had a major impact on ecosystem services.

As discussed further below, archaeological data, palaeontological records and historical documents provide numerous examples of human-driven declines in species abundance or shifts in species functional traits. In many cases, decimations and functional losses took place or were initiated long before we started collecting rigorous scientific data (e.g., Jackson et al., 2001; Lotze et al., 2006; Dulvy et al., 2009).

## Fidelity of geohistorical archives

The use of geohistorical archives to assess extinctions and extinction threats relies on the assumption that the subset of biota preserved in palaeontological, archaeological and other geohistorical records is an adequate and representative surrogate for all taxa. Numerous case studies and meta-analyses suggest that geohistorical archives provide meaningful estimates of key ecosystem properties, including diversity, community composition, relative abundances, food web structure or even spatial ecological gradients (e.g., Kidwell, 2001, 2007; Kidwell and Holland, 2002; Tomašových and Kidwell, 2009; Tyler and Kowalewski, 2017, 2023; Roopnarine and Dineen, 2018; Hyman et al., 2019; Pruden et al., 2021). In this context, it is useful to ask how accurately fossil archives would depict the conservation status of species included in the IUCN Red List database.

For the four major invertebrate taxa considered here (Figure 1), each species can be scored in terms of its fossilisation potential based on the presence and attributes of a preservable skeleton. Here, we classified all species (Supplementary Appendix 4) into three preservational categories: 0 – none or low (organisms with no biomineralised skeleton or with microscopic skeletal parts only), 1 – intermediate (organisms with weakly biomineralised skeletons or with multi-elemental skeleton prone to disarticulation), 2 – high (organisms with heavily biomineralised skeletons). The preservational potential categories were assigned at the "order" level, based on dominant skeletal type within a given group. Category 2 was used as "preservable taxa" in the analysis presented on Figure 3 (see Supplementary Material for additional details). These data can be used to evaluate if the current Red List assessment of all taxa is tracked adequately by the subset of taxa that could be recovered from the fossil record. We compared the distribution of species across the IUCN categories in the subset of taxa with a high fossilisation potential to the distribution based on the entire dataset (Figure 3). The analyses excluded brackish species because nearly all of them belong to the preservable category, and thus, the data subset is nearly identical to the entire dataset. For marine invertebrate species listed in the Red List, most species have a high potential of preservation (Figure 3a), which is not surprising given that most mollusc species and some arthropod and cnidarian groups have sturdy biomineralised skeletons and high fossilisation potential. Even for freshwater and terrestrial species, a substantial subset of taxa should be frequently preserved as fossils. Consequently, the preservable species are a robust predictor of all species when it comes to their conservation status, including marine (Figure 3b vs. 3e), freshwater (Figure 3c vs. 3f) and terrestrial (Figure 3d vs. 3g) systems. Note that the observed high fidelity is statistically inevitable: data subsets are expected to correlate with entire datasets unless a given subset is a very small portion of all data or represents a highly biased sample of that

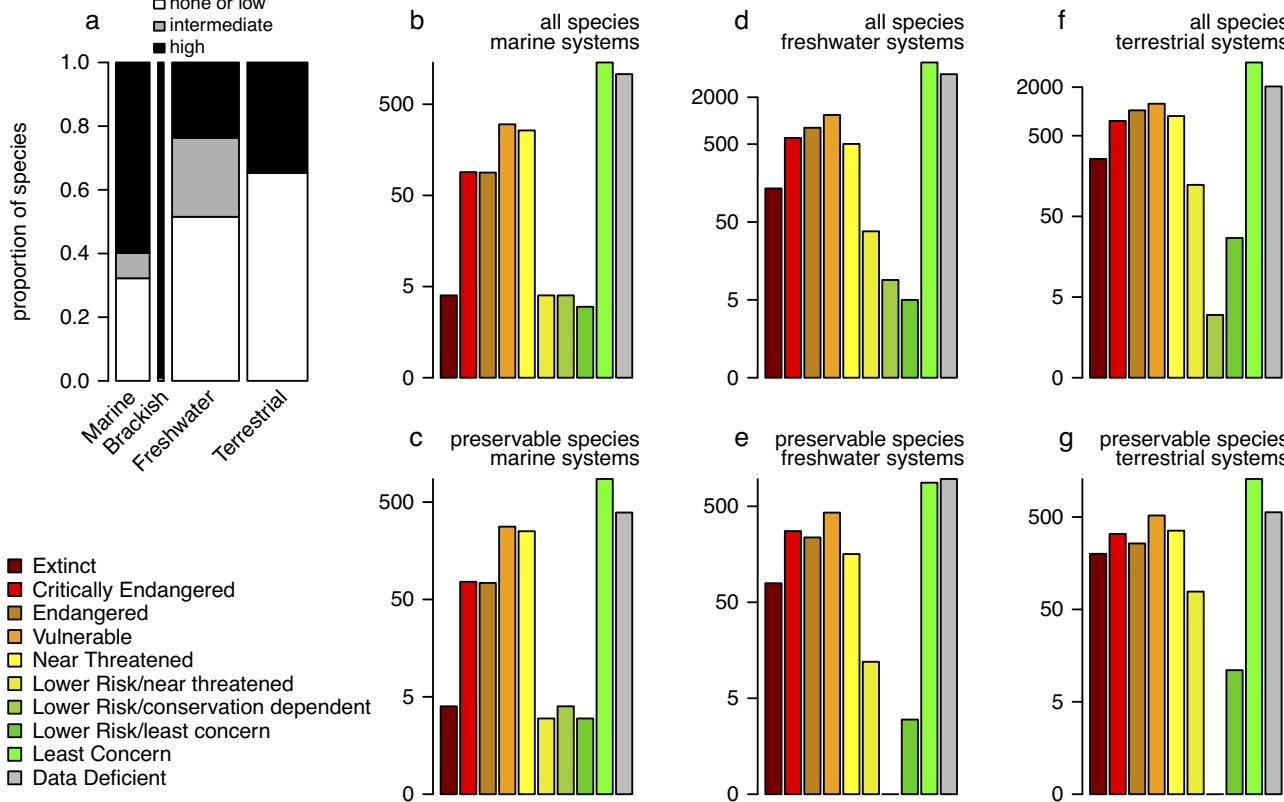

**Figure 3.** (a) Proportions of taxa with none/low, intermediate, and high preservation potential grouped into marine, brackish, freshwater, and terrestrial systems with data restricted to four invertebrate phyla (arthropods, cnidarians, echinoderms, and molluscs). The bar width is scaled to the square root of the total numbers of species (the numbers reported on Figure 1a). (b–f) The barplots of all IUCN-assessed species (b, d, and f) compared to barplots of a subset of those species that belonged to orders categorised as 'preservable' (c, e, and g). (b-c) Marine systems. (d-e) Freshwater systems. (f-g) Terrestrial systems. Different colours represent different IUCN conservation status categories. Brackish systems were excluded because almost all brackish species assessed by IUCN are preservable so the two plots would be virtually identical. All data from the IUCN Red List Database were accessed on 2/20/2023 (IUCN, 2023a).

dataset. However, this is the gist of the argument: in most cases, a preservable subset of taxa is fractionally large enough and unbiased enough to be expected to represent a reasonable proxy of the whole.

Whereas taxonomic fidelity of fossil archives is likely to be high, geohistorical samples tend to be affected by temporal mixing (time averaging; reviewed by Kidwell, 2013; Kidwell and Tomašových, 2013; Tomašových et al., 2023). As the rates with which skeletal remains are permanently buried is slow relative to the generation times of most organisms, remains of individuals that died at different times tend to accumulate on and within the seabed and can be mixed with older hardparts exhumed from deeper sediment layers by bioturbation or physical reworking. Due to time averaging, even the most finely resolved fossil samples may contain specimens that lived decades, centuries or even millennia apart, and the scale and structure of temporal mixing may vary across sediment layers, core segments or taxa (e.g., Kowalewski et al., 1998, 2018; Kosnik et al., 2009, 2015; Scarponi et al., 2013; Terry and Novak, 2015; Nawrot et al., 2022). In addition to reducing temporal resolution, time averaging can produce spurious patterns in the fossil record by increasing sample-level diversity, decreasing compositional turnover between fossil assemblages and obscuring abrupt regime shifts (Kidwell and Tomašových, 2013; Tomašových et al., 2020). Consequently, reconstructing historical patterns and processes can be challenging and often needs to be augmented by age dating of individual fossils so the data can be properly

interpreted (Tomašových et al., 2023). This will be further illustrated by multiple case studies discussed below.

Another challenge involves estimating population density and biomass from fossil archives. The abundance of fossils is controlled not only by productivity but also by net accumulation rates, intensity of mixing processes and the resulting time averaging (Kidwell, 1986). And whereas numerical estimates of population density are possible in certain cases (e.g., Kowalewski et al., 2000; Tomašových et al., 2019a), they usually require extensive age dating of individual specimens to "unmix" time-averaged assemblages, or samples from depositional settings characterised by exceptional temporal resolution (e.g., deep sea basins with high sedimentation rates). Similarly, translating skeletal estimates of body size into biomass can also be challenging because it is often difficult to establish a robust correlation between body size estimated from skeletal remains and the biomass of soft tissue (Powell and Stanton, 1985). Although such models have been developed for some groups, such as molluscs (e.g., Meadows, 2019) or fish (e.g., Granadeiro and Silva, 2000), they are frequently species-specific. Nevertheless, as illustrated in numerous examples below, these limitations can often be minimised, and robust numerical estimates can be derived from palaeontological and archaeological archives.

Finally, palaeontological archives available for sampling tend to be spatially and temporally discontinuous. However, the spatio-temporal coverage of those archives is still orders of magnitude better than the coverage provided by those few ecological

monitoring time series that extend back in time for more than just the last few decades. For example, few rigorous quantitative surveys of marine habitats have been conducted in the nineteenth and early twentieth centuries, resulting in a very spotty spatial knowledge of those relatively recent ecosystems. We cannot retrospectively survey the mostly unstudied nineteenth-century ecosystems, but palaeontological and archaeological archives are still accessible to sampling for many of those ecosystems and can serve as surrogate proxies for monitoring data for many regions and time intervals.

## Extinctions

### *Neontological knowledge of marine extinctions*

The International Union for Conservation of Nature (IUCN) is a reference standard, with assessments completed for 150,300 species (IUCN, 2023b). In the IUCN Red List, less than 1% of surveyed species are considered extinct in the wild (1,245 of over 150,388 examined; IUCN, 2023c, accessed April 26, 2023). If we consider "possibly extinct" taxa in the Red List, the magnitude does not change substantially (<2%, IUCN, 2023c). As mentioned above (see also Figure 1), most of the extinct species represent terrestrial or freshwater taxa. For the marine and brackish realms, Red List reports only 20 extinct species (Table 1), including 15 vertebrates (9 birds, 4 mammals and 2 fish species), 1 species of algae and, at most (see Cowie et al., 2022), 4 invertebrates (all 4 representing intertidal or brackish gastropods). Of these 20 extinct species, the larger vertebrates – for example, the iconic Steller's Sea Cow (*Hydrodamalis gigas*) – were rapidly driven to extinction due to

overhunting during the colonial period (Estes et al., 2016). In contrast, the only plant classified as extinct (the alga *Vanvoorstia bennettiana*) went extinct due to land use and pollution (Woinarski et al., 2019).

Notably, the extinctions reported so far are restricted to species that lived in coastal or brackish settings, which is not surprising given that coastal zones are the most severely impacted part of the marine realm (Halpern et al., 2008), are more easily accessible for inventory studies than offshore settings and have higher sampling coverage than continental margins or bathyal settings (O'Hara et al., 2020). But even in the case of coastal habitats, the human-driven extinctions primarily affected terrestrial dwellers of the coasts and islands, especially along the modern-time colonisation routes. For example, the ground doves from the Mascarene archipelago (the Dodo and its sister taxon, the Solitaire) were wiped out by European colonisers during the early modern time (e.g., Cheke, 2008; Cheke and Hume, 2008), but there are no records of any marine extinctions from the same region. This information is consistent with the notion that marine extinctions substantially lag terrestrial extinctions (Dulvy et al., 2009; McCauley et al., 2015). Alternatively, however, the lag may reflect the dearth of conservation assessments in the marine realm (Webb and Mindel, 2015). That is, we may have been much better at documenting the demise of ground doves and butterflies than sponges and snails.

Although the recent fossil record provides data on marine species that went extinct in the late Quaternary (e.g., the flightless marine duck *Chendytes lawi* from California; Jones et al., 2021 or shearwater species from the Canary Islands mention above), and thus co-existed with early human populations, the actual causes of

**Table 1.** The list of marine species currently classified as 'extinct' according to the IUCN Red List (IUCN, 2023b)

| Species | Class | Common name |
|---|---|---|
| *Vanvoorstia bennettiana* | Floridoephyceae | Seaweed |
| *Neomonachus tropicalis* | Mammalia | Caribbean Monk Seal |
| *Zalophus japonicus* | Mammalia | Japanese Sea Lion |
| *Hydrodamalis gigas* | Mammalia | Steller's Sea Cow |
| *Neovison macrodon* | Mammalia | Sea Mink |
| *Camptorhynchus labradorius* | Aves | Labrador Duck |
| *Prosobonia cancellata* | Aves | Christmas Sandpiper |
| *Bulweria bifax* | Aves | Small Saint |
| *Urile perspicillatus* | Aves | Spectacled Cormorant |
| *Zapornia monasa* | Aves | Kosrae Crake |
| *Pterodroma rupinarum* | Aves | Large Saint Helena Petrel |
| *Haematopus meadewaldoi* | Aves | Canarian Oystercatcher |
| *Pinguinus impennis* | Aves | Great Auk |
| *Mergus australis* | Aves | Merganser |
| *Prototroctes oxyrhynchus* | Actinopterygii | New Zealand Grayling |
| *Psephurus gladius* | Actinopterygii | Chinese Paddlefish |
| *Collisella edmitchelli*[a] | Gastropoda | Limpet |
| *Lottia alveus* | Gastropoda | Eelgrass Limpet |
| *Omphalotropis plicosa* | Gastropoda | — |
| *Littoraria flammea* | Gastropoda | Periwinkle |

[a]*Lottia edmitchelli* is currently recognised as the valid name for this limpet species.

the disappearance of those species may be difficult to discern, making the role of humans uncertain. These interpretative challenges potentially affect multiple species of marine invertebrates that went extinct in the late Quaternary, including, for example, the clam *Coanicardita californica,* the whelk *Pusio fortis* or the limpet *Lottia edmitchelli* (Harnik et al., 2012; Cowie et al., 2022). In fact, Cowie et al. (2022) argued that there is only one well-documented case of human-driven extinction in marine invertebrates (*Lottia alveus*).

### Empty shells: A hidden record of Holocene extinctions?

Empty shells of molluscs have been an important source of bio-diversity data, as they are routinely included in taxonomic studies and diversity surveys at local and regional scales (e.g., Mikkelsen and Bieler, 2000; Bouchet et al., 2002; Warwick and Light, 2002; Bieler and Mikkelsen, 2004; Zuschin and Oliver, 2005). According to Mikkelsen (2011), the majority of new modern bivalve species named between 2000 and 2009 were described from empty shells. Moreover, a survey of shelled marine gastropod species reported in 2006 revealed that 80% of species descriptions were restricted to shell morphology (Bouchet and Strong, 2010). Even when a region is studied over a longer time period, the significance of empty shells for estimating diversity remains impressively high. After 25 years of intensive exploration in New Caledonia, as many as 73% of 1,409 turrid gastropod species were only documented by empty shells, and 34% were known from a single specimen (Bouchet et al., 2009).

Are all those species known only from empty shells still around? Or are all those species extinct or extirpated, and, if so, what were the causes of their demise and when exactly did they disappear? These alternative explanations are difficult to resolve because many of these species are extremely rare. Nevertheless, the possibility that at least some of the species known only from empty shells are now extinct, or at least locally extirpated, cannot be ruled out (see also Diamond, 1987), especially given that even well-preserved shells accumulating on the seafloor can be hundreds to thousands of years old (e.g., Flessa and Kowalewski, 1994; Kidwell, 1998; Dexter et al., 2014; Butler et al., 2020; Ritter et al., 2023; Tomašových et al., 2023). To our knowledge, the data on the "empty shell species" are not being systematically collected, and specimens representing those species have not been subject to radiocarbon dating, which could potentially provide useful chronological constraints for those taxa. Currently, it remains unclear whether the prevalence of "empty shell species" in mollusc biodiversity studies reflects a long tail of rare species or represents the yet unacknowledged record of hidden extinctions and extirpations.

Whereas new species descriptions and occurrence records based on "empty shells" are pervasive among molluscs, they are unlikely to occur among many other common groups with biomineralised skeletons. Many of those taxa, such as stony corals and echinoids, are much less diverse than molluscs, so it is less likely that echinoid tests or coral fragments encountered on modern seafloor may represent unknown species. Also, unlike molluscs, for which many species have been defined based on shell characters only, many other marine taxa require soft tissue analysis for species-level identification. This requirement makes it less likely that a new species could be erected based solely on skeletal remains or that they would be used in monitoring surveys to establish the presence of a species. In the case of corals, the further limitation stems from the fact that the diagnostic features of corallites degrade rapidly after death (Greenstein and Pandolfi, 1997).

However, molluscs may not be the only group affected by the "empty shells" syndrome. Some subsets of benthic foraminifera and brachiopod species were also identified from dead material only (Logan et al., 2008; Milker and Schmiedl, 2012). Murray (2007) estimated that dead-only species of benthic foraminifers may have represented ~5% out of the ~2,140 documented species. Moreover, even when the type material of many species is based on live-collected individuals, the present-day species-level distribution maps and estimates of geographic or bathymetric range sizes are frequently based on the combination of live-collected individuals and dead-collected skeletal remains, including surveys of molluscs (e.g., Dijkstra and Maestrati, 2010), brachiopods (e.g., Bitner and Logan, 2016) and bryozoans (e.g., Di Martino and Rosso, 2021). The use of dead individuals in recent surveys may overestimate the present-day geographic or bathymetric ranges of marine species and thus underestimate the frequency of extirpations or ecological extinctions.

### Deep-time approaches

Although this review deals with near-time approaches that focus on the late Quaternary transition of the marine biosphere from pre-human to human times, the deep-time fossil record can also be useful in assessing or predicting modern extinctions and extinction threats. In the geological past, short periods of rapid global warming and acidification ($10^3$ year) are increasingly employed as ancient analogs of near-future outcomes (e.g., Kiessling et al., 2023). The biological record of these deep-time events can inform us about the most widespread processes that may be driving extinctions during hyperthermal events (e.g., Benton, 2018; Foster et al., 2018). For example, assessing simulated impacts of global warming on marine invertebrates against empirical patterns recovered from the fossil record of several deep-time hyperthermal events suggests that ongoing warming has the potential to annihilate endemic taxa in cold-water habitats within a single century (Reddin et al., 2022).

Other deep-time approaches have focused on assessing extinction rates and extinction selectivity. In particular, the background extinction rates estimated from the fossil record have been used as a benchmark for assessing if recent species are disappearing at an abnormally high pace (e.g., Pimm et al., 1995; Barnosky et al., 2011; Harnik et al., 2012; Lamkin and Miller, 2016; Cowie et al., 2002). In addition, the spatial and taxonomic selectivities of past extinctions have been used as predictors of extinction vulnerability for present-day species and habitats (e.g., Harnik, 2011; Harnik et al., 2012; Finnegan et al., 2015; Collins et al., 2018). Finally, the deep fossil record has been used to test the predictive power of species-area relationships (SAR) models for estimating extinction rates due to habitat loss (see Preston, 1962; Rybicki and Hanski, 2013). For example, in the Pliocene succession of San Joaquin (California), SAR model predictions for biodiversity shifts, expected due to sea-level changes, underestimated the species loss observed in the fossil record (Pruden and Leighton, 2018).

### Extirpations

A complete extirpation of a species from a given region is difficult to prove because once the species is rare, it would hardly be recorded anymore. A recent reassessment of the IUCN Red List indicated that overfishing drove over one-third of all sharks and rays towards global extinction (Dulvy et al., 2021), but while such commercial extinctions are well documented, the complete disappearance from

a given region is rarely certain. For example, following the collapse of the once economically important angel shark *Squatina squatina* in the northern Adriatic Sea, the species was never caught in scientific surveys. However, fishermen reported that the species was still observed but rarely (Fortibuoni et al., 2016). Although geohistorical data cannot assist with those challenges, they can be invaluable by identifying extirpation events that occurred before systematic bio-inventorying efforts started.

Unknown or poorly known extirpation events can be detected using data derived from archaeological middens, the late Quaternary fossil record or ancient DNA. These methods can be particularly effective when used jointly. For example, archaeological and ancient DNA data demonstrated that both right and grey whales occurred in the Strait of Gibraltar region during the Roman period and that grey whales still occurred along the Asturian coast during pre-Roman times (Rodrigues et al., 2018). These data document extirpation events that would remain unknown if our knowledge were to be derived from neontological data alone.

Geohistorical data can also aid in assessing the potential role of humans in driving extirpation events. For example, combined use of ancient DNA and radiocarbon dating revealed that the genetic diversity of Atlantic grey whale, restricted today to the North Pacific, declined gradually in the mid-Holocene long before the onset of intensive commercial whaling, indicating that this extirpation event was likely precipitated by Holocene climate changes or other ecological causes (Alter et al., 2015). This is in contrast to the case of right whales, which were the main whaling target in the North Atlantic until becoming commercially extinct in the mid-eighteenth century (Rodrigues et al., 2018).

The disappearance of less charismatic animals widely preserved in the fossil and archaeological record can be readily documented by geohistorical data (e.g., benthic mollusc shells, fish otoliths and bones). For example, data derived from historical records, archaeological middens, death assemblages (i.e., surface accumulations of skeletal remains) and radiometrically dated sediment cores demonstrate that oyster reefs underwent extirpation in the late nineteenth and twentieth centuries in many temperate regions, including the coast of Victoria, Australia (Ford and Hamer, 2016), Tasmania (Edgar and Samson, 2004), eastern Scotland (Thurstan et al., 2013) and the northeastern Adriatic Sea (Gallmetzer et al., 2019). Similarly, an analysis of death assemblages revealed that epifaunal suspension feeders (scallops, brachiopods) were abundant on the southern California mainland shelf during the late Holocene (with standing density of at least 20 individuals/m$^2$) but were subsequently extirpated (except for shelf-edge relic populations), most likely, due to the nineteenth-century increase in sedimentation and turbidity induced by agricultural land use (Tomašových and Kidwell, 2017).

Geohistorical insights not only can help us to detect extirpation events but can also be used to reconstruct shifts in functional traits and life history characteristics of species with rapidly declining populations, thus providing baseline data needed to improve the management of such species during restoration efforts. For example, sclerochronological analyses of prehistoric otoliths revealed major changes in growth rate and maturation time of an endangered marine fish *Totoaba macdonaldi*, endemic to the Gulf of California, caused by upstream diversions of the Colorado River flow (Rowell et al., 2008), illustrating the value of such historical approaches for revising our understanding of the ecology of endangered species now only represented by remnant populations.

On occasions, especially when aided by geochronological age dating, geohistorical data can refute human-induced stressors as a cause of extirpation. For example, age dating of valves of the semelid deposit-feeder bivalve *Ervilia purpurea* in the Persian Gulf implied a boom-and-bust population dynamics, suggesting that its current absence in the living assemblage of the region is unlikely to be linked to the onset of oil platform production in the twentieth century (Albano et al., 2016).

In summary, a growing body of literature demonstrates that conservation palaeobiology approaches not only allow us to detect unknown extirpation events and provide information for species and ecosystem management but also make it possible to assess the role that human activities may have played in driving those events.

## Ecological extinctions

### Decimations

In contrast to extinctions or extirpations that are difficult to detect conclusively, geohistorical archives provide direct records of local or regional population size trajectories of formerly abundant species that became decimated. And given that the late Quaternary fossil record of marine environments is globally widespread and well-resolved stratigraphically, it can provide an impressive spatio-temporal coverage of formerly abundant organisms that started to decline decades, centuries or even millennia before rigorous bio-inventorying efforts ensued.

For example, sedimentary cores collected across multiple regions of the northern Adriatic Sea documented that over the past two centuries, multiple, formerly abundant suspension-feeding or herbivorous molluscs declined in abundance due to trawling, pollution and eutrophication (Gallmetzer et al., 2019; Tomašových et al., 2019a, 2020). In fact, a regional shell bed, formed by shells of mollusc species that were decimated during the nineteenth and early twentieth century, is still present just below the seafloor across large portions of the NE Adriatic shelf (Gallmetzer et al., 2019; Tomašových et al., 2019a). This is a forceful testament to a highly diverse regional benthic ecosystem that perished before we started assembling a rigorous scientific knowledge of the region's seafloor. These major regional changes to benthic ecosystems could not have been detected based on biomonitoring surveys, which only started in the twentieth century.

In many cases, geohistorical studies not only document species declines that predate modern biomonitoring efforts but can provide estimates of the natural range of variability, which can then be used to gauge the significance of human-induced decimations. For example, the decline in the diversity and percent cover of reef corals induced by pollution, heat stress, overfishing and acidification are well documented (Jackson et al., 2001; Pandolfi et al., 2003; Aronson and Precht, 2006; Precht et al., 2020). But how do they compare to natural variability in coral cover? After all, Holocene-scale studies document significant declines in abundance and carbonate production of corals over the past millennia that were unrelated to anthropogenic impacts and driven by climatic and sea-level fluctuations (Perry and Smithers, 2010; Toth et al., 2012, 2018; Yan et al., 2019; Leonard et al., 2020). These natural changes can serve as a benchmark to demonstrate that the magnitude and extent of losses of coral habitats and their diversity driven by human activities do typically exceed the natural range of variability (Pandolfi and Jackson, 2006; Cybulski et al., 2020; O'Dea et al., 2020; see also Cramer et al., 2017, Muraoka et al., 2022). Similarly, sedimentary cores from the coastal Adriatic habitats indicated that shifts in mollusc communities during the ice ages over the last 125,000 years were much less dramatic than changes in relative

species abundance that took place in the last centuries (Kowalewski et al., 2015). These data also demonstrated that those mollusc communities were spectacularly resilient to major climate and sea-level changes in the late Quaternary (Kowalewski et al. 2015; Scarponi et al., 2022) but not to late Holocene human impacts (Scarponi et al., 2023). These examples highlight the unique value of geohistorical estimates in assessing if a given human-induced ecosystem shift is a truly significant event or falls within the natural range of long-term ecosystem variability.

Even for species with low preservation potential, long-term population dynamics can often be inferred from indirect proxies, especially if such species modify their environment in a way that leaves strong signatures in the sedimentary record. For example, geochemical biomarkers such as sterols and stable nitrogen isotopes ($\delta^{15}N$) derived from bird guano and preserved in coastal pond sediments can be used to track shifts in colony size of nesting seabirds and seaducks (e.g., Hargan et al., 2019; Duda et al., 2020). By applying this approach to dated lake sediment cores, Duda et al. (2020) demonstrated that the world's largest colony of a threatened Leach's Storm-petrel (*Hydrobates leucorhous*; Baccalieu Island, Canada) was smaller than today and fluctuated in size for most of its 1,700-year history, putting recent declines observed since the 1980s in a broader historical context. Lake sediment records of nitrogen isotopes and other geochemical proxies have also been used to reconstruct centennial-scale changes in population size of anadromous fish such as sockeye salmon (*Oncorhynchus nerka*) and link those changes to climate and fishing pressures (Finney et al., 2000).

### Relative estimates of decimations

It is instructive to examine specific strategies used to quantify decimations and identify ecological extinctions. In general, geohistorical studies compare living communities with either surficial death assemblages or Holocene records from cores and outcrops (e.g., Kidwell, 2007; Kowalewski et al., 2015; Albano et al., 2016; Hyman et al., 2019; Sander et al., 2021). These efforts are often supplemented with radiometric dating, stable isotope analyses or ancient DNA sampling (e.g., Kowalewski et al., 2000; Sivan et al., 2006; Tomašových et al., 2019a; Dillon et al., 2021). In addition, some studies also combine fossil and archaeological records to detect formerly abundant or habitat-forming species known to be rare or absent today (Rick et al., 2016; Fariñas-Franco et al., 2018).

When using geohistorical records, decimations can be inferred indirectly by measuring the decline in the relative abundance of a species in a series of palaeontological samples (Figure 2c). This approach is straightforward to implement in practice and can show that a given taxon declined in ecological importance relative to other taxa but does not provide numerical estimates of pre-impact population size or average density – information that may be crucial for guiding restoration efforts.

Despite those limitations, changes in relative abundance among preservable marine taxa not only provide records of ecological extinctions predating modern bio-inventorying but can also potentially reveal selective decimations that preferentially affected certain functional groups and shifted communities into new functional states (Kidwell, 2008; Steger et al., 2021). For example, the youngest stratigraphic record indicates that, during the last two centuries, the species sensitive to pollution or hypoxia (including foraminifera, ostracods, molluscs and corals) declined in abundance and geographic extent, while those that were tolerant to various stresses concurrently increased in dominance (Gooday et al., 2009). These

patterns were observed in many regions of the world, primarily based on data from sediment cores collected in river-dominated coastal environments. The non-exhaustive examples include hypoxia-related changes in (1) benthic foraminifera from the Louisiana shelf (Blackwelder et al., 1996; Osterman et al., 2005; Platon et al., 2005), the North Sea (Polovodova et al., 2011; Dolven et al., 2013; Nordberg et al., 2017), the Tagus Delta (Bartels-Jónsdóttir, 2006) and the Gulf of St. Lawrence (Thibodeau et al., 2006; Genovesi et al., 2011); (2) ostracods from the Chesapeake Bay (Cronin and Vann, 2003); (3) molluscs from the Gulf of Trieste (Tomašových et al., 2020); and (4) corals from China's Greater Bay Area (Cybulski et al., 2020). Whereas many of these habitats were also exposed to eutrophication or oxygen depletion due to natural climatic variability over the past millennia, the magnitude of the resulting ecosystem changes was typically much less pronounced when compared to changes induced by recent anthropogenic impacts (Cooper and Brush, 1993; Osterman et al., 2009; Li et al., 2011).

### Numerical estimates of decimations

In contrast to relative assessment, numerical estimates of decimation provide direct estimates of the decline in abundance, often estimated comparatively as changes in the population density (number of specimens per unit of area) or other units that can be simultaneously measured for modern and fossil taxa. Such numerical estimates are much more informative than relative assessments but are much more challenging to derive and require multiple assumptions that can be partly constrained by age dating and by other methods (species lifespan estimates, rates of disintegration of skeletal remains in the surface layer, and net sediment accumulation rate; Tomašových et al., 2023). Nevertheless, multiple examples of numerical assessments have been published over the last two decades, demonstrating that these strategies are feasible and can provide quantitative estimates of changes in marine populations. These strategies tend to be idiosyncratic, being tailored to unique aspects of each case study. And even though those estimates tend to be approximate, they allow us to detect major ecosystem changes. Moreover, in cases of major shifts in species abundance, often by multiple orders of magnitude, the somewhat elevated imprecision of numerical estimates derived from geohistorical archives is typically inconsequential. It is also noteworthy that numerical estimates are typically derived in a maximally conservative manner (e.g., Kowalewski et al., 2000).

One of the earliest direct estimates was derived for benthic ecosystems of the Colorado River delta, which was drastically altered due to the construction of numerous dams in the upstream parts of the river (Fradkin, 1996). A combination of field surveys, field sampling, numerical dating and oxygen isotope analyses of shell material provided a strategy for estimating the past population density of benthic molluscs (Kowalewski et al., 2000). Using the maximally conservative estimates that yielded minimised estimates of past population density, geohistorical data suggested that during the last millennium, the intertidal population density averaged at least 50 adult molluscs $m^{-2}$. In contrast, the surveys of the modern intertidal zone yielded an estimate of 3 molluscs $m^{-2}$, suggesting an almost 20-fold decline in mollusc abundance. These data also indicated that restoration efforts did not bring the local benthic productivity back to its pre-industrial levels. Subsequent conservation palaeobiology studies in the delta area also demonstrated that geohistorical approaches can be used to estimate how the shutdown of the river affected water flow (Dettman et al., 2004), life history of

aquatic organisms (Rowell et al., 2008), predation processes (Cintra-Buenrostro et al., 2005) and net carbon emission (Smith et al., 2016).

Similarly, Lockwood and Mann (2019) compared the density of living oyster populations from the Chesapeake Bay to fossil populations of the Pleistocene age. However, due to time averaging, live and fossil populations were not directly comparable because Pleistocene shells of oysters occurring together in situ likely record a mix of multiple generations, thus providing misleadingly high estimates of standing population density. Because dating methods available for Pleistocene deposits do not offer a sufficient resolution to correct for time averaging, dead–live ratios in modern oyster reefs were used to derive adjusted (and highly conservative) estimates of Pleistocene population densities. The resulting density estimates for live oysters in the Pleistocene record were an order of magnitude higher than those obtained for modern oyster populations from the same area. The Pleistocene estimates also notably exceeded the threshold density of 50 oysters $m^{-2}$ used in Chesapeake Bay as a benchmark for a fully recovered oyster population.

Age dating of skeletal remains provides key information on parameters needed to reconstruct population density because age-frequency distributions are informative about disintegration rates, net sediment accumulation rates and time averaging. For example, Tomašových et al. (2017) investigated whether high densities of the opportunistic, hypoxia-tolerant bivalve *Varicorbula gibba* – induced by eutrophication in the northern Adriatic Sea during the late twentieth century – were novel or had analogs over the past 500 years. Taking into account the disintegration rate of bivalve remains and net sediment accumulation rate (estimated on the basis of age model), assuming a maximum lifespan equal to 5 yr. in a core with cross-sectional area of 0.04 $m^2$, they estimated that maxima in abundances of this species correspond to a standing density of 1,250–1,500 individuals/$m^2$, a density similar to times of *V. gibba* outbreaks observed today. In contrast, radiometric age dating revealed that one of the major contributors to carbonate sands in the northern Adriatic Sea, the bivalve *Gouldia minima*, which was abundant in the last few thousand years, declined to almost zero abundance over the past two centuries due to the anthropogenically driven loss of algal and seagrass meadows (Tomašových et al., 2019a). A similar approach used to infer past population densities also detected unusually high densities of the deposit-feeding bivalve *Nuculana taphria* on the southern California shelf during the Holocene, followed by a two-order-of-magnitude decline in its abundance during the twentieth century (Tomašových et al., 2019b).

A different approach relies on using accumulation rates of skeletal elements, estimated based on core age models, as a proxy for species abundance. For example, Dillon et al. (2021) compared shark denticle assemblages from a mid-Holocene Caribbean reef with those found in modern death assemblages. In this case, Uranium-Thorium and calibrated radiocarbon dating of coral pieces were used to estimate the time encompassed by the sediment samples and calculate reef accretion rates. The denticle accumulation rates standardised for reef accretion rates suggested that sharks were over three times more numerous before humans began using marine resources in Caribbean Panama. Similar strategy was used to document historical declines in sea urchin (Cramer et al., 2017) and parrotfish populations (Muraoka et al., 2022) on Caribbean coral reefs, as well as long-term fluctuations in pelagic fish populations based on scale and otolith deposition rates (e.g., Field et al., 2009; Finney et al., 2010; Jones and Checkley, 2019).

In addition to direct numerical estimates, the magnitude and timing of population decline in the past can be estimated using numerical age dating. For example, dating revealed that the bivalve *Glycymeris nummaria* appeared in the Eastern Mediterranean in large numbers 5,000–5,500 years ago and almost ceased to exist 1,500–1,000 years ago, probably due to the ongoing impoverishment of nutrient flux and reduction in marine productivity when the sea level rise in the late Holocene slowed down and reached modern levels (Sivan et al., 2006).

The above examples suggest that numerical estimates of decimations as well as the timing of proliferation and decimation events can be estimated from geohistorical records, given assumptions about lifespan, disintegration of skeletal remains and net sediment accumulation rate. These declines can assist us in identifying extinction threats that have deep historical roots.

### Shift in functional traits

A decline in ecosystem services provided by a species can occur not only due to decimation but also because of shifts in its functional traits (Figure 2c). The most common functional losses involve demographic changes, which can result in the loss of large-size classes and older or more reproductively active age cohorts. Although such changes are rarely invoked in the context of ecological extinctions, in size-structured populations, different life stages or age classes can interact with different subsets of species in a community and play different ecological roles, and thus, their selective removal may lead to functional loss (Ebenmman et al., 2017). For instance, experimental evidence suggests that deep-burrowing adult stages of large, long-lived bivalves provide key ecosystem functions in soft-sediment habitats but take years to recover following local disturbances such as seasonal hypoxia (Norkko et al., 2013). Decrease in body size and other life-history changes induced by fishing may shift the ecological niches and functional roles of harvested species, destabilising food webs and potentially triggering trophic cascades (Hočevar and Kuparinen, 2021).

Life-history changes can be inferred using geohistorical approaches by surveying size frequency distributions of fossil populations and by examining growth rates and longevity, which can be assessed using sclerochronological approaches (e.g., Goodwin et al., 2001; Rowell et al., 2008; Lockwood and Mann, 2019). In addition, the fossil and archaeological records provide numerous examples of studies documenting major shifts in functional traits that could be linked to human activities, especially selective harvesting (e.g., Limburg et al., 2008; O'Dea et al., 2014; Rick et al., 2016; Ruga et al., 2019; Assumpção et al., 2022; reviewed by Sullivan et al., 2017). Whereas in many cases human activities resulted in shorter lifespans and slower growth rates of marine organisms, this was not always the case. For example, archaeological data suggest that the construction of clam gardens (intertidal rock-walled terraces) by indigenous people resulted in the increased growth rates and size at the time of death of maricultured clams (e.g., Toniello et al., 2019).

The Chesapeake oyster study mentioned above (Lockwood and Mann, 2019) provides a forceful case example in which the Pleistocene fossil record was used to show that populations of oysters in the past included a higher proportion of large individuals, with the largest size classes notably exceeding the largest live oysters observed today. Integration of data on population density and demography of Pleistocene oysters suggested that filtration rates for those populations were an order of magnitude higher than those

estimated for modern populations (Lockwood and Mann, 2019), thus providing direct estimates of the decline in ecosystem services due to shifts in functional traits.

## Concluding remarks

The review of the marine conservation palaeobiology literature demonstrates the potential of geohistorical approaches for assessing recent extinctions and extinction threats while also highlighting the strengths and limitations of those approaches.

Firstly, the existing literature demonstrates that, despite spatial and temporal gaps, geohistorical archives provide a comprehensive spatial and environmental coverage of marine systems at coarser observational scales. That is, the palaeontological and archaeological samples can be acquired for many regions and habitats of the world. This is particularly valuable in those areas that either lack any past ecological surveys or have been surveyed only in the last few decades. In all such regions, skeletal remains are likely to exist on the seafloor and should allow for bio-inventorying of taxa that were common in the area in the last centuries or millennia.

Secondly, the conservation palaeobiology studies demonstrate that geohistorical approaches are applicable to many groups of organisms, including molluscs, corals, ostracods, foraminifera, fish and marine mammals, to list just a few examples. Moreover, they represent a substantial fraction of all taxa and often can serve as surrogate proxies for the entire communities to which they belong (e.g., Tyler and Kowalewski, 2017, 2023; Kokesh et al., 2022; and references therein). This is important because taxa with an excellent fossil record, such as molluscs, may help elucidate biodiversity dynamics in marine ecosystems during and before the early modern times. And whereas the biodiversity losses following human migrations are well documented for conspicuous, iconic taxa (e.g., megafaunal extinctions in North America; Meltzer, 2020), a more comprehensive understanding that encompasses all taxa remains elusive (see also Cowie et al., 2022).

Thirdly, rapid advances in dating techniques and instrumentation allow for dating smaller aliquots at a faster pace and lower costs, making it feasible to date hundreds of specimens in single projects. Age dating of shells or bones will continue to uncover extinctions and extirpations in the recent past and help us to assess if humans may have played a significant role in those events. And whereas conservation palaeobiology studies often encounter difficulties in determining the human role in past extinction events, the age distributions of dated specimens can potentially estimate the precise timing of extinctions and extirpation events and provide numerical assessments of decimations, which in turn can help us to identify extinction threats. The literature also suggests that geohistorical archives are a great resource for understanding the recent past and identifying human-driven changes that have already occurred but would be difficult to elucidate without palaeontological or archaeological data.

Finally, most geohistorical studies, including many examples highlighted in this review, indicate that many ecosystems have deteriorated in terms of taxonomic and functional diversity, spatial range and continuity, and functional ecology of individual species. Those geohistorical data indicate that marine ecosystems have been accumulating a human-driven extinction debt for centuries or even millennia.

In summary, despite various limitations and assumptions that underlie conservation palaeobiology strategies, geohistorical archives represent a wealth of data that complement ecological and conservation efforts and will likely continue to play an important role in assessing extinctions, extirpations, ecological extinctions, extinction debts and extinction threats.

## Future research directions

Conservation palaeobiology is a relatively new research direction so the trivial notion that we need more case studies is germane here. This is especially so for the marine realm, for which only a few groups of organisms, most notably molluscs and corals, have been studied more extensively using geohistorical approaches. And even in the case of molluscs or corals, the geohistorical coverage is still limited and primarily focused on coastal systems. However, given the rapid growth of conservation palaeobiology research, we expect that new case studies will be added at an accelerating pace. In addition to the obvious need for more case studies across regions, ecosystem types and organismal groups, several research themes are particularly noteworthy.

- Global scale meta-analyses – Currently, there are too few case studies for any marine ecosystem type or any fine-scale groups (e.g., genera, families) of marine organisms to allow for any robust meta-analyses on global or multi-regional scales (with a notable exception of a live-dead meta-analysis of benthic molluscs, see Kidwell, 2008). However, with new case studies being added every year, there is a good prospect that such larger-scale comparative analyses will become feasible in the foreseeable future. Already there exist multiple geohistorical case studies focused on closely related marine species making it possible to seek common patterns and processes, as in the case of declines in abundance of acroporid corals in the Caribbean (Cramer et al., 2020) and Great Barrier Reef (Clark et al., 2017). In both regions, the decline started decades before the onset of monitoring efforts in the 1970s and 1980s.

- Geochronology – There is a steady increase in the number of studies that use age dating (especially U-Th, $^{14}$C), and these dating methods are becoming increasingly affordable and require smaller aliquots thus allowing for dating smaller specimens (e.g., Bright et al., 2021). The dating of large samples of marine skeletal remains is needed for many systems and groups of organisms to better understand the temporal resolution and coverage of geohistorical data (Zuschin, 2023). For example, recent efforts to date echinoids yielded disparate estimates of time averaging: subdecadal for large sand dollars (Kowalewski et al., 2018) but multi-centennial for minute clypeasteroids (Nawrot et al., 2022). The echinoid conundrum illustrates the need for extensive dating across taxa and depositional systems to develop a more robust understanding of the temporal resolution of geohistorical data.

- "Empty shell" species – Recent studies on molluscs and foraminifera suggest that some unknown fraction of the present-day marine species were described from skeletal remains of organisms. Given that many skeletal remains can reside on seafloors for centuries or millennia, the "empty shell" species may alternatively represent a record of rare extant species, species that went extinct due to natural processes, or species that disappeared due to human activities. Studies that would focus on understanding how pervasive are "empty shell" species for various groups of marine organisms, as well as projects focused on dating those species to assess their time distribution, could advance our understanding of the present-day biodiversity in the marine realm and improve estimates of recent extinctions.

- Terrestrial–marine transitions - In coastal areas, targeted geohistorical research in the marine realm could be used to compare marine and terrestrial extinctions and extirpations for the same coastal system. Exploring marine extinctions in regions where early terrestrial extinctions attributed to humans have been already documented – as in the example of the ground doves from the Mascarene archipelago wiped out by colonisers during the early modern time – would be particularly fruitful.

- Translating research into action – The biggest challenge of conservation palaeobiology revolves around practical applications of geohistorical data. This is a two-pronged issue of being able to translate scientific knowledge into appropriate conservation actions and understanding what type of geohistorical data would be most useful to practitioners (Dietl et al., 2019; Kiessling et al., 2019). This issue is not specific to extinction-focused studies but any geohistorical studies that aim to inform conservation efforts. Whereas translating research into conservation actions is beyond the scope of this review, it should be considered explicitly in any studies that aim to use historical data to assist present-day conservation efforts.

**Open peer review.** To view the open peer review materials for this article, please visit http://doi.org/10.1017/ext.2023.22.

**Supplementary material.** The supplementary material for this article can be found at http://doi.org/10.1017/ext.2023.22.

**Data availability statement.** No new data are reported in this review. The IUCN Red List dataset used to generate Figures 1 and 3 are reposited on GitHub, and access links are provided in Supplementary Material.

**Acknowledgements.** We thank the IUCN community for developing, maintaining, and allowing access to the Red List database, a subset of which has been used here to generate Figures 1 and 3. We thank three anonymous reviewers, Jonathan Payne (Associate Editor) and John Alroy (Editor) for constructive comments that substantial improved this review.

**Author contribution.** Conceptualisation: M.K., R.N., D.S. A.T., M.Z.; Writing —original draft: M.K., R.N., D.S., A.T., M.Z.; Writing—review and editing: M.K., R.N., D.S., A.T., M.Z.

**Financial support.** The project benefited from resources provided by the Conservation Paleobiology Network funded by the US National Science Foundation grant EAR 1922562. MK thanks the University of Florida Foundation Thompson Fund for partial support of this project. AT thanks the Slovak Research and Development Agency (APVV 22–0523) and the Slovak Scientific Grant Agency (VEGA 02/0106/23) for funding.

**Competing interest.** The authors declare no competing interests exist.

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
