## [Reviewer Report]

This brief manuscript provides a clear and succinct review of several ways that Quaternary fossil records are used to detect recent biotic change, including, global extinctions (in principle), and regional extirpations, functional shifts, and so on. If published, this manuscript would join a number of reviews (including a couple authored by some of the authors of this manuscript) that highlight the applications of young fossil records in conservation (e.g., Dietl & Flessa 2011; Kidwell 2013; Kidwell & Tomasovych 2013; Dietl et al. 2015; Kosnik & Kowalewski 2016). These reviews differ to varying extent in their focus, depth, and audience. Some, such as Kidwell (2013), review the recent conservation paleo and geochronological literature for an audience primarily consisting of other conservation paleo researchers. This is evidenced in part by their publication in paleo journals (e.g., Palaeontology), but also because they consider in detail the complexities (nuances?) of geohistorical data and methods, and outline unresolved questions that could serve as a research agenda for others already in the field. Other reviews (e.g., Kosnik & Kowalewski published in Biology Letters) are intended primarily for conservation biologists who may be less familiar with geohistorical data and methods. As such, these reviews are outreach efforts that review concepts and data that are increasingly well established among paleobiologists but largely unfamiliar to neontologists. Reviews for both of these audiences are important. While an effective review ideally provides all readers with new perspectives and guiding questions for future work, the center of gravity undoubtedly will depend on whether the audience is primarily paleo or neo in focus. This manuscript by Kowalewski and others, primarily leans towards the latter (i.e., neontological audience), revisiting the paucity of data on recent marine extinctions and current extinction risk (see also Harnik et al. 2012 TREE, Webb & Mindel 2015 Current Biol); the spatiotemporal limitations of biological monitoring and real time instrument data (e.g., Kosnik & Kowalewski 2016). These introductory sections provide a motivation for the main focus of the review (Ms. pgs. 6 – 17), but depending on the audience may be content that is already well established (e.g., neo readers may be familiar with the dearth of IUCN risk assessments for marine taxa; paleo readers with the limitations of direct observation data; neo and paleo readers familiar with extinction terminology). My primary over-arching comment is that the manuscript could be strengthened by being more deliberate about the intended audience(s) and the content most relevant for that audience. If neo for example, some of the content above could be omitted or reduced and relevant information that is largely absent in the current Ms (e.g., the spatiotemporal grain of geohistorical data), could be provided.

With respect to figure 1, the authors’ main message is clearly conveyed but I was curious why they chose to group the finer-grained IUCN risk assignments into these four broader categories. The only way to understand the correspondence between the grouping used in the figure and the IUCN assignments, is to review the code in the Supplementary Materials. At minimum, the mapping between these two sets of classification should be made explicit in the main text. The authors may want to consider though whether simply using the IUCN assignments would be clearer.

Figure 2 is a nice schematic intended for a neontological audience on the potential contributions of geohistorical data for understanding extinctions, extirpations, and ecological extinctions. A strength of the associated caption is that it highlights specific case studies reviewed in the text for each of these general patterns.

Figure 3 and associated analysis are unclear as presented and additional information is needed prior to publication. I understand the authors’ intent - i.e., to understand whether the conservation status of preservable taxa can serve as a surrogate for all taxa. However, insufficient information is provided for how species were assigned to preservation categories (i.e., Fig3a), and what is being plotted in Fig 3b-3e is unclear. With respect to preservation assignments, what criteria were used (e.g., biomineralized? Multi-element? Identifiability? Mineralogy?) and at what taxonomic level were these preservation assignments made? Were these binary (yes/no), or multi-state measures (e.g., weakly biomineralized)? Were these assignments ground-truthed against the fossil record and, if so, how? With respect to Figs 3b-3e, I am unsure what is being graphed. The caption states that these panels illustrate the relationships between the conservation status of preservable taxa vs. the conservation status of all taxa. However, conservation status is not plotted here as far as I can tell; although it could be if each point were color coded according to mean conservation status and a legend was provided. I am assuming that each of the points is a higher taxon of some sort, and that the x and y axes are the number of preservable species vs total species richness, respectively, but what are these subclades? Are they all at the same taxonomic rank? How were these assignments made and which clades are included or excluded in these panels? More detail is needed. Also, f the authors are regressing species richness for a subset of the data against species richness estimates for all the data, isn’t this a regression-to-the-mean problem, which might result in the high r-squared values reported in each panel? More details are needed in order to assess this analysis and its correspondence with the statements made in the main text.

A few additional minor comments for the authors to consider:

1. The spatial patchiness of biomonitoring data are contrasted with the potentially global scope of geohistorical data. While true in principle, in practice geohistorical data are also spatially patchy and consequently sampling and analytical methods that consider spatiotemporal resolution and extent are essential for analysis of both of these data types.

2. On pg. 6 the authors indicate that extinction debts and relaxation times are challenging to measure but suggestion that integration of paleo and neo data and modeling efforts could help. How specifically? Additional information is needed here in order to understand how this potential goes beyond “more data (or different types of data) might help us to address this problem.”

3. On pg. 9 the authors state that marine extinctions may substantially lag behind terrestrial extinctions. To what extent is this perception exacerbated by the paucity of biological monitoring of marine species? Consider for example Webb & Mindel 2015 (Current Biology) who showed that apparently lower risk of extinction among marine taxa reflected in part limited risk assessments of marine species. Perhaps marine extinctions lag terrestrial extinctions globally, but if Webb and Mindel are correct, extirpations and ecological extinctions are roughly comparable between the marine and terrestrial realm (when comparing similarly-studied groups)? Webb and Mindel (2015) may also be worth considering at the end of the Introduction when discussing Figure 1a and 1b.

4. On pg. 15, the authors state the numerical assessments of decimations “have high cognitive value.” I am unsure what this means. Does this mean that relatively simple models for deriving numerical abundance estimates can be informative with respect to order of magnitude shifts in abundance over time, despite being incorrect in detail? Consider rewording.

5. On pg. 15 in discussing Colorado River restoration efforts (“restoration efforts had not returned the local benthic productivity to its pre-industrial levels”) are there studies that have been published since Kowalewski et al. 2000 that provide additional data on the efficacy of restoration efforts? I wonder whether any of the work by Jansen Smith & Greg Dietl would be relevant here.

---

## [Reviewer Report]

Summary: This is a succinct, well-written review paper that highlights recent work in marine conservation paleobiology. It is timely, building on the work of the Conservation Paleobiology Network, and very up-to-date in terms of the published literature it emphasizes. I’ve recommended a few revisions, including: (1) broadening the taxonomic and habitat breadth of the case studies, (2) providing a clearer description of Figure 3 and its implications, and (3) adding a section on future research directions.

Key points

- I didn’t get a chance to count your case studies, but they seemed heavy on molluscan and European examples. It would be great to add a bit more taxonomic and habitat breadth if possible.

- Figure 1 is a super useful comparison. Are there data available on both a and b through time? For each environment? It would be fascinating to plot the equivalent of a “collector’s curve” for these data. This would also allow you to potentially predict how far along marine sampling is, relative to other environments.

- Figure 3 is quite difficult to follow, especially for a biological or ecological audience. What does each dot represent on the graphs? Is it species within families?

- It might be beyond your purview, but it would be really useful to include a Future Directions section in this review. This section could highlight which directions you think are most promising. Right now, the review ends a little weakly with summary points.

General

- Do you need “the” before the phrase “geohistorical” throughout the paper?

- Standardize spelling of mollusc (or mollusk) throughout

Abstract

- You mention geohistorical records at the global scale, but I didn’t see many global case studies mentioned in the text. How useful are global (as opposed to local or regional) paleontological studies for marine conservation?

Impact Statement

- You mention that “…whereas modern scientific data and strategies will remain the most critical dimension of biodiversity conservation and ecosystem management, the geohistorical approaches can augment those efforts further…” But you don’t really expand on this in the text. Are there examples of geohistorical data that have actually been used for management? Can you discuss most effective practices for doing this?

- “…losses that already took place” to “…losses that have already taken place…”

Introduction

- You mention that “modern scientific data inform us about the most recent eco-environmental changes in selected regions, even though those changes have been…happening all over the world.” Isn’t this an issue for the fossil record too? To what extent are global paleontological studies actually global?

- The paragraph that begins “Over the last several decades…” Does the RCN have a website of publications that could be cited here? As proof of concept and to highlight the RCN website as a source for readers?

Extinction Terminology

- This section is an excellent addition to the review—especially given the possible breadth of your readership.

- “Yet the concept of extinction debt is valuable in its scalable potential…” Not sure what you mean here. Can you reword and explain in more detail? Do you return to the concept of extinction debt in the body of the review?

Extinction

Neontological knowledge

- “…not surprising, given that coastal zones are the most severely impacted part of the marine realm…” What role, if any, does sampling bias play in this pattern? Presumably, coastal zones are better known, better sampled, etc. than deeper habitats.

Empty shells: A hidden record

- Last paragraph: Really interesting point—and nicely explained. Would this be the case for other taxa, for example corals?

Deep time approaches

- Not sure that this section adds much—It’s so abbreviated that you don’t have much of a chance to review the literature. Also not sure how well it fits with the previous section.

Ecological extinctions

- It is worth adding 1-2 sentences re. the challenge of estimating biomass in the fossil record—for your neontological audience who may wonder why it’s rarely used in paleontological studies

Shift in functional traits

- Paragraph “Life history changes” could definitely be expanded upon to include more examples, especially with respect to sclerochronology

Table 1

- Cormoran should be Cormorant?

- Nmerganser should be Merganser?

- Standardize capitalization

Figure 1

- Typo in caption (Red LIst)

Figure 2

- Labels for C seem a little vague or confusing positioned—especially functional traits.

- Italicize genus and species names in caption

---

## [Editor Report]

I have received two thoughtful reviews of this manuscript. The reviews are largely concordant in indicating that the manuscript is well written, of broad interest, and should be published following appropriate revisions. It is particularly strong in showing the value of the near-time fossil record for understanding natural versus anthropogenic variation and for establishing a true, pre-human baseline. The latter is a particular challenge for marine systems, where modern scientific records address only the most recent century or two in systems that have been altered by human activities for millennia.

I read the manuscript myself and concur with nearly all the reviewer comments and so will not add my own. Among these comments, I think the most salient are (i) for the authors to consider the comment regarding intended audience and whether the section of definitions of terms is necessary or could be revised and (ii) revised figure 3, which did not appear to me to correspond exactly to the figure caption, explaining the confusion from the reviewers.

I look forward to receiving a revised manuscript along with a point-by-point response to the reviewer comments.

---

## [Reviewer Report]

REVIEW TEXT

The authors did a great job responding to the majority of the comments from the editor and two reviewers. The revised manuscript is considerably improved. I still have a few concerns—primarily with respect to Figure 3 and the discussion of spatial patchiness (outlined below).

FIGURE 3

The revision of Figure 3 does make it a bit clearer, especially with the addition of the supporting legend and SOM text. But I’m still struggling with what the take-home message is--- presumably it’s the idea that the preservable subsample of IUCN species in each habitat (marine, brackish, etc.) yields the same distribution of species across IUCN categories. If that’s the case, surely there’s an easier way to show this in a figure?

By presenting the comparison as a diagonal line—it suggests that data points should be distributed on both sides of the line. But the lower right corner of the graph won’t (by definition) contain any data points. And the r2, plotted with the 1:1 line, suggests that the data points are being compared statistically to the 1:1 line—but that isn’t the case.

If you’re trying to show that the preservable subset is representative of the whole population, then isn’t the distance between each point and the 1:1 line the important metric? That could be shown much more simply (bar graph?)—and the distances could be quantitatively compared across IUCN categories to see if they vary significantly. Another option would be to compare the slope of the preservable subset to the slope of the whole population.

Either way, I’m not clear on why the current r2 is useful?

With the legend added, I’d like to see a discussion of how the patterns vary according to IUCN category (i.e., symbol shape and color). For example, why do the higher risk categories plot in the center of the line—with the lower risk categories plotted on the lower and higher ends of the line?

If the authors choose to keep the figure, it could be improved with explicit mention of: (1) the source of the taxonomic information (used to categorize species into orders), (2) the source of the preservational information, (3) the source of the total number of species, (4) what the width of the bars in Fig 3A mean. I am also wondering whether there is a more appropriate metric for comparing set to subset than a regression?

FIGURE 1

Have the authors considered trying to bootstrap the sample size of “Terrestrial” and “Freshwater” down to the sample size of “Brackish” and “Marine” to see how it affects the ratios of the IUCN categories?

GRAPHICAL FIGURE

The authors did add a graphical abstract— but I don’t think it adds much to the paper. It’s the same figure as Fig 2. One or the other could be deleted to streamline the paper.

TEXT

I think I understand what the authors are arguing, re. the patchiness of the neontological versus the paleontological record. But the argument still feels a little simplistic. If I understand correctly, the authors are arguing that because neontological sampling can only happen now (i.e., over recent decades), it will, by definition, be more poorly sampled geographically than paleontological data, which can be sampled at any time in the future. But I don’t actually think that’s true—for a few reasons. First, the stratigraphic record preserves some habitats much more reliably than others. Some habitats are almost never preserved. Second, the stratigraphic record, once deposited, can always be destroyed, by erosional, tectonic, geochemical, anthropogenic, and other processes. Third, fossils and sediment can be mixed, both spatially and temporally, for both neontological and paleontological sampling. Do we expect the magnitude and duration of the mixing to differ from neo—versus paleo? If we are arguing that extinction debt can be defined temporally, is there an equivalent in geographic space? In addition, the authors seem to be contradicting themselves re. modern versus fossil record patchiness on p. 10 versus 23 of the manuscript.

I definitely agree with retaining the definition section and would suggest: (1) changing the title to Terminology and (2) adding the terms “relaxation time” and “time averaging” to it

MINOR TYPOS

- One mollusk left in Future Directions setting

- The phrase losses that “already took place”—still occurs in two other places in the text

- Typo in phrase “allowing to use geohistorical data for predicting future extinction risk”

---

## [Reviewer Report]

This is an interesting contribution focused on how the late Quaternary fossil record can inform our understanding of marine extinctions, extirpations, and ecological extinctions. I reviewed the initial manuscript (reviewer #1) and appreciate the edits made by the authors in this revised submission. Most of my remaining comments are relatively minor and focus on clarity (listed below), with the exception of a few points regarding the analysis underlying Figure 3, which I think necessitate revision prior to publication.

Regarding Figure 3:

- lines 40-50 on pg. 8 introduce the approach used to determine whether the relative risk status of species with high fossilization potential reflect risk assessments for full dataset in four different ecosystems (e.g., marine, freshwater). In this analysis, fossilization potential was scored relatively coarsely (i.e., at the ‘order’ level and using a tripartite scheme) and these details are currently subsumed in the supplement. I recommend these methods be moved to the main text where they will be evaluated by a greater number of readers engaging with the paper.

- I was confused by several of the fossilization classifications included in Appendix 4. Specifically, why were unionds assigned a “1” rather than a “2” like all other bivalve mollusks? Similarly, why were tryblidiids, a group of monoplochophoran mollusks assigned a “1” rather than a “2” like most of the other orders of calcifying mollusks? A few other categorizations I was unsure about included the barnacle ‘order’ Sessilia which was assigned a “0” in contrast with the statement in the supplement that discussed barnacles as a “2” because of their heavily biomineralised skeletons, and the contrasting scores for two orders of holothurians; ASPIDOCHIROTIDA which was assigned a “1” which seems high in contrast with the HOLOTHYROIDAE which was assigned a “0” which is more accurate reflection of biomineralization in this overall group). Because unionids are species-rich and include many at risk species, this categorization may qualitatively affect the results plotted in Fig 3a and 3d, although this will be modest due to the inevitable statistical association described the authors on pg 9 of the main text and in their randomisation analysis in the supplement.

- Figure S1 and associated text indicate that the high r-squared values observed in Figure 3 are expected because the preservable taxa are a subset of the entire dataset. Figure S1 could be strengthened by including the observed r-squared values in the respect randomisation plot for that ecosystem.

- The caption for Figure 3 should clarify that these data consist only of species assessed by the IUNC. In other words, “Total # of species” equals “Total # of assessed species,” which for many of these marine invertebrate groups may not correlate with total described species richness. Similarly, “# of preservable species” is limited to the subset of species assessed by the IUCN in those groups that has been characterized as having high preservation potential.

Additional recommendations to consider:

- Pg 3, lines 50-51: reword so that you don’t assert that all recent eco-environmental changes observed in select regions are manifest everywhere on the globe. Indeed, a strength of these geohistorical data is the opportunity to assess the extent that this statement is true.

- Pg 6, line 32: replace “premise” with “promise”

- Pg. 7, line 14-17: provide additional detail to clarify the timing of extirpation and range expansion in these two groups of penguins, similar to the example discussed in the preceding sentence

- Pg. 14, line 43: include reference for commercial extinction of the N. Atlantic right whale

- Pg. 14, lines 44-47: add “shells” after “benthic mollusc” and move this parenthetical content to end of sentence after “geohistorical data”

- Minor text issues (grammar, punctuation, spelling) occur occasionally throughout the manuscript and addressing them at this stage will streamline subsequent copy-editing.

Overall, this is an interesting synthesis of recent conservation paleo studies that are relevant for our understanding of extinction, extirpation, and ecological extinction. The authors have done a fine job of highlighting a diversity of case studies that span an array of organisms, regions, and methodological approaches. I anticipate that this review will be of interest to marine biologists studying both modern and paleo ecosystems and recommend it for publication pending revision.

---

## [Editor Report]

I received two thoughtful reviews of the revised manuscript, both from the original reviewers of the manuscript. These reviews agree that the revised manuscript is much improved and largely addresses the comments raised regarding the initial submission.

The reviewers provide a few additional comments and suggestions for further improvement. While I do not believe the manuscript requires further review beyond my own inspection, I would like to see the authors address the comments received when the submit the revised manuscript. I do not anticipate that this will be burdensome.

The one item where I do think revision is required is regarding Figure 3. One reviewer notes that the scatterplots in Figure 3 are not entirely appropriate to the data, given that values cannot plot to the bottom/right side of the 1:1 line because the preservable species are, by definition, a subset of the total species. Plotting the ratio of preservable to total species on the vertical axis vs taxonomic identity (or number of total species) on the horizontal axis would be more appropriate and is equivalent to the reviewer’s suggestion of looking at the distance of each point from the 1:1 line, given that the data are plotted on a logarithmic scale. I also agree with the reviewer that the R^2 values are not particularly informative because the data don’t meet the assumptions of linear regression.

I look forward to receiving a revised version of the manuscript.

---

## [Editor Report]

I am satisfied that the authors have addressed all of the reviewer comments and thank them for considering this final round of review. I recommend acceptance of the manuscript.